# Leaf Extracts of *Miconia albicans* (Sw.) Triana (Melastomataceae) Prevent the Feeding and Oviposition of *Plutella xylostella* (Linnaeus, 1758) (Lepidoptera: Plutellidae)

**Isabella Maria Pompeu Monteiro Padial** [1], **Silvana Aparecida de Souza** [2], **José Bruno Malaquias** [3], **Claudia Andrea Lima Cardoso** [4], **Jéssica Karina da Silva Pachú** [5], **Claudemir Antonio Garcia Fioratti** [2] **and Rosilda Mara Mussury** [2,*]

1. Faculty of Agricultural Sciences, Federal University of Grande Dourados, Highway Dourados-Itahum, km 12, Dourados 79804-970, Brazil
2. Faculty of Biological and Environmental Sciences, Federal University of Grande Dourados, Highway Dourados-Itahum, km 12, Dourados 79804-970, Brazil
3. Entomology Laboratory, Agrarian Science Center, Federal University of Paraíba, Areia 58397-000, Brazil
4. Center of Studies in Natural Resources, State University of Mato Grosso do Sul, Highway Dourados-Itahum, km 12, Dourados 79804-970, Brazil
5. Department of Entomology and Acarology, Luiz de Queiroz College of Agriculture (ESALQ), University of São Paulo (USP), Av. Pádua Dias 11, Piracicaba 13418-900, Brazil
* Correspondence: mussuryufgd@gmail.com

**Abstract:** Sustainability in food production is an increasingly discussed issue nowadays; therefore, demands for research that can reduce production costs and ensure the quality and autonomy of production are relevant, with attention to the use of plants due to their importance in biodiversity. Thus, the objective of this research was to evaluate the bioactivity, feeding preference, and oviposition preference of *Miconia albicans* botanical extracts at concentrations of 1%, 5%, and 10% against *Plutella xylostella*. We observed reduced larval duration, larval survival, female hatching success, and repellence of oviposition and feeding for all concentrations. For higher concentrations, the extracts showed a larval mortality rate of 58%, a feeding reduction of 82%, and an oviposition reduction of 94%, showing potential for pest control. Phytochemical analyses identified phenolic compounds, flavonoids, and tannins, which are substances with repellent and larvicidal properties. This is the first report on the phytosanitary potential of *M. albicans*, showing that the plant has both lethal and sublethal effects on *P. xylostella*.

**Keywords:** integrated pest management; botanical insecticide; diamondback moth; organic production

## 1. Introduction

Botanical insecticides are products extracted or derived from the secondary metabolites of plants that can be used to control agricultural pests [1], including *Plutella xylostella* (Linnaeus, 1758) (Lepidoptera: Plutellidae) [2–6]. Secondary metabolites are naturally produced as a plant defense strategy; however, there is still a lack of phytochemical, toxicological, and pharmacological studies for much of the known flora [7]. The effect of the product is directly related to the time of collection, plant species, organ collected, general plant condition at the time of collection, and method of preparation of the extract [8].

Currently, pest mortality is considered the most relevant effect of botanical insecticides, along with their lower environmental damage. However, nonlethal effects are also observed, such as feeding-repelling [9], oviposition-repelling [5,6,10], and growth-blocking effects [4]. Botanical insecticides also show potential for harm reduction in agricultural plantations [11].

Pest management using alternative control methods, specifically plant extracts, has been studied to minimize the impact of pesticides [12]. This is because plant extracts are

less toxic, typically contain several bioactive compounds, and have low environmental persistence and a generally low cost of use, being particularly suitable for small farmers with limited income [13].

One option fulfilling the abovementioned conditions is the prospection of medicinal plants, which has raised great interest due to the possibility of discovering new bioactive compounds to originate, for example, plant-based insecticides. Plant-based insecticides act synergistically, presenting attractive, dislodging, or repellent features and phagoinhibitory action, in addition to containing substances capable of altering the growth and development of some insects [14–16]. Many of these substances are produced in response to insect attacks [17] and thus are employed in the control and monitoring of insect pests [18].

The family Melastomataceae belongs to the order Myrtales, consisting of 170 genera and approximately 4500 species [19]. *Miconia*, the largest genus in the family, comprises approximately 250 species occurring in Brazil [20,21]; however, little is known regarding its phytochemical potential.

*Miconia albicans* (Sw.) Triana (Melastomataceae) is one of the most important species in the Melastomataceae family and is locally known as "folha-branca" or "canela-de-velho". Traditionally, it is used to treat arthritis, rheumatism, arthrosis, vitiligo, and genitourinary infections, as well as to prevent heart attack, regulate heart rhythm, relieve febrile symptoms and gastric problems, and as an antivenom [22,23]. Previous chemical studies have found compounds such as triterpenes, coumarins, tannins, flavonoids, and saponins in the plant [21]. Some of the cited substances have documented repellent, larvicidal, and fungicidal action, such as triterpenes [24,25] and flavonoids [26,27]; therefore, this study tested the hypothesis that the aqueous extracts of *M. albicans* have insecticidal potential against agricultural pests.

*Plutella xylostella* (Linnaeus, 1758) (Lepidoptera: Plutellidae) is a moth with a high spread capability and is able to adapt from temperate to tropical climates, making it a global pest [28,29]. The species is currently the main agricultural pest in more than 100 countries, and its annual management costs add up to USD 4-5 billion in losses worldwide [23,30]. *P. xylostella* is stimulated by the presence of glusinolates, affecting the species of the *Brassica* genus, such as cauliflower, broccoli, and collard greens [23,31]. Their larvae make leaf mines and defoliate, causing damage throughout the year except during rainy seasons [32,33].

Furthermore, *P. xylostella* populations are capable of developing resistance to pesticides in a shorter time due to their morphophysiological characteristics; as a consequence, the species have been reported to be resistant to 101 active ingredients [34–41]. The toxicity and pesticide resistance development of this pest have limited its control; therefore, the development of alternative management methods based on biological pesticides is particularly important for the control of this insect [42].

The agricultural market is increasingly pressured by a very large demand for new phytosanitary products every year, more specifically, for insecticides and alternatives that can maintain the ecological balance of the agroecosystem and effectively control target insects. Therefore, the objective of this study was to (1) analyze the potentially lethal and sublethal effects caused by the botanical extracts of *M. albicans* leaves at concentrations of 1.0%, 5.0%, and 10.0% on the life cycle, and the feeding and oviposition preference of *P. xylostella*; and (2) to conduct a chemical analysis of the *M. albicans* extracts, in order to identify the classes of compounds possibly involved in larvicidal, anti-feeding, and anti-oviposition effects.

## 2. Materials and Methods

### 2.1. Preparation of Botanical Extracts

The botanical material was collected in the Lagoa Grande settlement in Itahum (22°05′ S and 55°15′ W), Mato Grosso do Sul. Fresh leaves of *M. albicans* were identified and deposited in the Herbarium of UFGD. The collection of botanical material was authorized by the Brazilian National Research Council (CNPq)/Council for Genetic Heritage Management (CGEN/MMA) under number AD06DBA.

Extract preparations were conducted at the Insect–Plant Interaction Laboratory (LIIP) of the Federal University of Grande Dourados (UFGD), School of Biological and Environmental Sciences, Dourados, Mato Grosso do Sul, Brazil.

The methodology for the botanical extract preparation was adapted from Ferreira et al. [9,43]. *M. albicans* leaves were sanitized using a 10% solution of sodium hypochlorite and dried in a forced-air oven at 45 °C for 3 days. The dried leaves were then ground using a knife mill, and the powder was stored away from light in plastic containers until the experiments began.

During the experiments, the botanical extract preparation started every day, one day before use. The plant powder was mixed with distilled water through the maceration technique at concentrations of 10% (3.0 g/30 mL), 5% (1.5 g/30 mL), and 1% (0.3 g/30 mL). The solutions were then stored in glass containers in a refrigerator, and after 24 h, they were filtered through filter paper so the final solution could be used for disc treatment.

### 2.2. P. xylostella Stock Culture

To obtain the initial individuals, larvae, and pupae of *P. xylostella*, the samples were collected from vegetable gardens in Dourados and Itaporã, Mato Grosso do Sul. *P. xylostella* rearing was carried out at the Insect–Plant Interaction Laboratory (LIIP) of the Federal University of Grande Dourados (UFGD), School of Biological and Environmental Sciences, Dourados, Mato Grosso do Sul, Brazil. The laboratory conditions were maintained as follows: temperature of 25 ± 1 °C, relative humidity 70 ± 5%, and photoperiod of 12 h. The rearing method was adapted from Barros et al. [44].

The *P. xylostella* larvae were kept in plastic cages (30 × 15 × 12 cm), which contained a sheet of paper towel for moisture absorption and two collard green leaves, arranged with the abaxial side facing each other. Every two days, the cage was sanitized with alcohol, the oldest leaf was replaced, and the newest leaf was passed down. The larvae remained in the cages until the pupal stage, when they were collected and transferred to another container.

The new pupae were taken to adult plastic cages (9 × 19 × 19 cm). These cages contained three collard green discs, three filter paper discs (both with 4 cm² Ø)—one on top of the other—and cotton wool soaked in a water and honey solution (10%). The adults that emerged remained in the cage until the end of their life cycle. The collard green discs, filter paper discs, and cotton were replaced every two days.

The filter paper and collard green discs containing eggs were then transferred to the larvae cages and removed after the eggs hatched and the neonate larvae moved to the collard green leaves.

### 2.3. Effects of M. albicans Extract on P. xylostella

All experiments (Figure 1) were conducted at the Insect–Plant Interaction Laboratory (LIIP) of the Federal University of Grande Dourados (UFGD), School of Biological and Environmental Sciences, Dourados, Mato Grosso do Sul, Brazil.

#### 2.3.1. Experiment 1: Bioactivity of Botanical Extracts against P. xylostella

The bioactivity of *M. albicans* experiments was analyzed based on the methodology of Peres et al. [4]. Neonate larvae were removed from the stock culture and placed individually in Petri dishes (4 cm diameter) containing a filter paper disc and two treated collard green discs, both measuring 4 cm². The treated discs were replaced daily, and the larvae were observed until they reached the pupal stage.

The larval duration (in number of days) and larval survival (number of individuals that reached the pupal stage) were evaluated.

Individual pupae were weighed 24 h after pupation (pupal weight) using a Bel Mark analytical balance with 0.001 g precision and monitored until the adults emerged (pupal survival).

The emerged adults were sexed, and the age-matched couples were transferred to individual cages (8 cm × 8 cm). Each cage contained a filter paper disc, a collard green

disc (both 8 cm in diameter), and cotton wool soaked in a solution of honey and water (10 g/100 mL), for adult feeding. Hatching success was evaluated daily, the collard green discs were replaced by new ones, and the eggs per couple were counted (number of eggs/day). The oviposition period was calculated based on the dates when the female started and stopped oviposition. Hatching success (percentage of hatched larvae) was assessed by counting the number of larvae at 4, 5, and 6 days after laying.

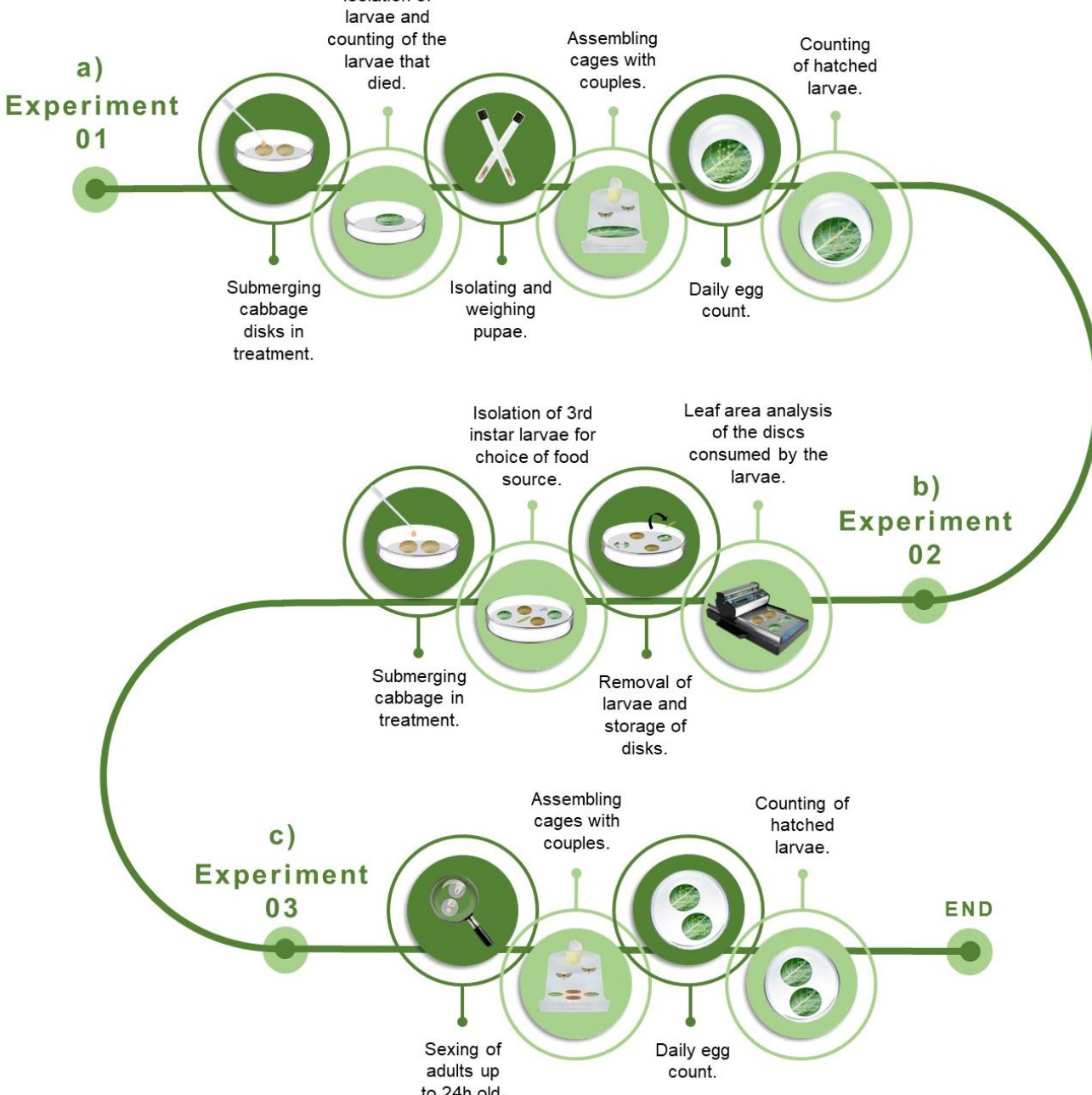

**Figure 1.** Schematic and chronological representation of the methodology used to evaluate lethal and sublethal effects of *M. albicans* botanical extracts on *P. xylostella*: (**a**) Experiment 1: bioactivity of botanical extracts against *P. xylostella*; (**b**) Experiment 2: free-choice feeding preference in *P. xylostella*; (**c**) Experiment 3: free-choice oviposition preference in *P. xylostella*.

The adults were monitored until they died (female and male longevity).

The bioactivity tests were based on the method of Peres et al. [4] (Figure 1a). The parameters evaluated were larval and pupal duration and survival, pupal weight, the longevity of females and males, and egg viability.

For the experiments, 50 replicates were used per treatment, in a total of 4 treatments comprising the extracts with concentrations of 1%, 5%, and 10%, and a control treatment (distilled water).

The experiments were conducted in a completely randomized design (CRD) with four treatments, one of which was the control.

### 2.3.2. Experiment 2: Free-Choice Feeding Preference Test with *P. xylostella*

The methodology for the free-choice feeding preference experiments was adapted from Kogan and Goeden [45]. The base of previously sterilized Petri dishes (8 cm$^2$) was lined with a filter paper disc (8 cm$^2$). Four collard green discs, two discs treated with the extract, and two discs treated with distilled water (4 cm$^2$ in diameter each) were placed crosswise and equidistantly onto the plate. Then, the 3rd instar larvae of *P. xylostella* were placed in the center of the plate.

The larvae remained in contact with the discs for 24 h and were then removed. The discs removed were scanned on the same day. The images generated by the scanner were analyzed using the ImageJ software [46] in order to calculate the total disc area and the area consumed in each disc. The parameters evaluated were the total disc area and the area consumed by the larvae (Figure 1b); the difference between these values was identified as leaf intake.

### 2.3.3. Experiment 3: Free-Choice Oviposition Preference Test with *P. xylostella*

The methodology for the free-choice oviposition preference experiments was adapted from Huang and Renwick [47]. The pupae of *P. xylostella* were removed from the stock culture and placed in test tubes until emergence. Adults aged up to 24 h were sexed, and couples were placed in plastic cages (8 cm × 8 cm). Inside the cages, there was a filter paper disc and four collard green discs (4 cm$^2$), two of them treated with extract and two with the control treatment. The discs were placed in an equidistant and intercalated manner. The adults were fed with a solution of honey diluted in water.

The discs containing eggs were removed daily during the first 10 days of female oviposition and replaced by new ones. The eggs were counted and stored in Petri dishes. After 5 days, the number of larvae was also counted. The parameters evaluated were the number of eggs and the number of larvae (Figure 1c).

### 2.4. Chemical Analyses of M. albicans Extracts

The chemical analyses were conducted at the Center of Studies in Natural Resources, State University of Mato Grosso do Sul (UEMS), Dourados, Mato Grosso do Sul, Brazil. All the extract concentrations were prepared according to Section 2.1. Each one of the concentrations was evaluated following the methodologies presented below. In the preparation of the blank, distilled water was used instead of the extract, the other reagents were inserted, and the reading was performed using a spectrophotometer (Global Trade Technology) in all tests. The experiments were performed in triplicate.

### 2.4.1. Phenolic Content Using the Folin–Ciocalteu Method

To 0.1 mL of the extract, 0.5 mL of the Folin–Ciocalteu reagent (1:10 $v/v$) and 1 mL of distilled water were added, and the sample was incubated for 1 min. Subsequently, 1.5 mL of 20% sodium carbonate ($w/v$) was added, and the mixture was allowed to react for 2 h, keeping the sample in the dark. Finally, the sample was read using a UV/Vis spectrophotometer at a wavelength of 760 nm [48].

For quantification, an analytical curve of gallic acid subjected to the same chemical reaction was used. The results were expressed as gallic acid equivalents per mL of sample (mg GAE mL$^{-1}$).

### 2.4.2. Flavonoid Content Using the Aluminum Chloride Method

To 1 mL of the extract, 1 mL of 2% aluminum chloride in methanol was added, and the mixture was allowed to react for 15 min and read in a UV/Vis spectrophotometer at a wavelength of 430 nm [48]. An analytical curve of rutin was prepared for quantification, and the results were expressed as rutin equivalents per mL of extract (mg RE mL$^{-1}$).

### 2.4.3. Tannin Content by Folin-Denis Method

The tannin content was determined using the Folin–Denis spectrophotometric method described by Pansera et al. [49], with modifications to the volumes of the reagents used, while maintaining the concentrations. To 0.5 mL of the sample, 0.5 mL of Folin–Denis reagent was added, followed by 0.5 mL of 8% sodium carbonate, and the mixture was allowed to react for 120 min. The reading was performed in a spectrophotometer at a wavelength of 725 nm. To calculate the tannin concentration, an analytical curve was performed using tannic acid standards. The results were expressed as mg of tannic acid equivalents per mL of extract (mg TAE mL$^{-1}$).

### 2.4.4. Antioxidant Potential by the DPPH Method

To 0.1 mL of the sample, 3 mL of the DPPH radical was added, and the reaction was incubated for 30 min in the dark. After this period, the sample was read at a wavelength of 517 nm. The percentage of DPPH inhibition (%) was calculated as described by Kumaran and Joel Karunakaran [50].

### *2.5. Statistical Analyses*

### 2.5.1. Experiment 1: Bioactivity of Botanical Extracts

All data were analyzed using deviance analysis. For the variables larval and pupal duration and pupal weight, no goodness of fit was found for the models tested (normal or gamma distribution with inverse link, identity, log, or quadratic functions). Therefore, these variables were analyzed with a nonparametric Kruskal–Wallis statistical test. In contrast, both female and male longevity showed a normal (Gaussian) distribution.

The generalized linear model with binomial distribution and overdispersion had goodness of fit for the larval survival and egg survival data, while the binomial model without overdispersion had the best goodness of fit for the pupal survival data. The goodness of fit of the aforementioned models was assessed with a half-normal plot [51]. The contrasts of the mean infestation level between the treatments were performed using the glht function of the multcomp package [52].

### 2.5.2. Experiment 2: Free-Choice Feeding Preference Test

The experiments were performed in a completely randomized design (CRD), with 4 treatments—1%, 5%, and 10%, and the control (distilled water)—with 30 replicates per treatment. The data were subjected to the Shapiro–Wilk normality test. Since the values did not follow a normal distribution, the leaf area consumed was compared between the treatments using the Kruskal–Wallis test ($p \leq 0.05$) and within each treatment using the Mann–Whitney test ($p \leq 0.05$).

The feeding deterrent effect was evaluated using the feeding preference index (FPI) of Kogan and Goeden [45]. The index classifies a substance as a feeding stimulant if the index is greater than 1, neutral if equal to 1, and a feeding deterrent if less than 1, using the following formula:

$$FPI = 2A/(M + A), \tag{1}$$

where
    A = the consumed area of the treated discs;
    M = the consumed area of the untreated disc.

2.5.3. Experiment 3: Free-Choice Oviposition Preference Test

The experiments were performed in a completely randomized design (CRD), with 4 treatments—1%, 5%, and 10%, and the control (distilled water)—with 10 replicates per treatment. The number of eggs deposited was compared between and within each treatment. The data distribution was not normal; therefore, the data were analyzed using contrasts from the quasipoisson generalized linear model ($p \leq 0.05$).

To calculate the oviposition suppression index (OSI), the formula by Kogan and Goeden [45] was adopted. When the OSI is greater than 1, the extract is a stimulant, and when the OSI is lower than 1, the extract is a deterrent. If OSI is equal to 1, it is neutral:

$$OSI = 2A/(M + A),\qquad(2)$$

where

A = the number of eggs in the leaves treated with the extract;
M = the number of eggs in the leaf treated with water.

**3. Results**

*3.1. Bioactivity of Botanical Extracts against P. xylostella*

The larval duration differed significantly from the control for all the extract concentrations evaluated. The extracts prolonged larval duration ($\chi^2 = 41.56$, *df* = 3, $p < 0.00001$) at the concentrations of 5% and 10%, and these values did not differ from each other, while the concentration of 1% reduced larval duration ($\chi^2 = 41.56$, *df* = 3, $p < 0.00001$) almost twofold compared with the control (Table 1).

**Table 1.** Duration (mean $\pm$ SE) and nonparametric ranking of the larval and pupal stages of *P. xylostella* exposed to different *M. albicans* extract concentrations.

| Concentration (%) | Larval (Days) | (Rank KW) | Pupal (Days) | (Rank KW) | Pupal Biomass (mg) | (Rank KW) |
|---|---|---|---|---|---|---|
| Control | 9.43 ± 0.31 | (80.00 b) | 5.22 ± 0.15 | (61.40 a) | 0.0038 ± 18.12 × 10⁻⁵ | (66.11 a) |
| 1.00 | 5.59 ± 00.14 | (21.82 c) | 5.11 ± 0.18 | (59.44 a) | 0.0039 ± 16.29 × 10⁻⁵ | (71.75 a) |
| 5.00 | 10.40 ± 00.50 | (93.66 a) | 5.08 ± 0.15 | (56.02 a) | 0.0036 ± 53.01 × 10⁻⁵ | (58.06 a) |
| 10.00 | 9.57 ± 00.63 | (78.90 c) | 5.38 ± 0.20 | (65.97 a) | 0.0036 ± 16.72 × 10⁻⁵ | (58.55 a) |
| *p*-value | <0.00001 | | =0.8051 | | =0.4382 | |

KW rank: Nonparametric Kruskal–Wallis ranking; *p*-value: probability value. Means followed by the same lowercase letter in the column did not differ significantly based on the Kruskal–Wallis test at 5% probability.

The survival ($\chi^2 = 39.51$, *df* = 3, $p < 0.0001$) of larvae treated with all concentrations of the botanical extract was reduced. The 10% concentration caused the lowest survival rates; however, it did not differ significantly from the 5% concentration (Table 2). Larval survival gradually decreased as the extract concentration increased, eliminating up to 58% of the *P. xylostella* population. Egg survival ($\chi^2 = 13.94$, *df* = 3, $p < 0.0001$) was lower for the 1% concentration, with a reduction of 11.76% compared with the control (Table 2).

**Table 2.** Parametric ranking using the binomial model with survival (mean $\pm$ SE) of the *P. xylostella* larval and pupal stages and egg viability at different extract concentrations of *M. albicans*.

| Concentration (%) | Larval Viability (%) | | Pupal Viability (%) | | Eggs Viability (%) | |
|---|---|---|---|---|---|---|
| Control | 96.00 ± 02.79 | a | 95.74 ± 02.97 | a | 91.95 ± 01.24 | a |
| 1.00 | 74.00 ± 06.26 | b | 91.89 ± 04.54 | a | 81.13 ± 03.51 | b |
| 5.00 | 50.00 ± 07.14 | c | 92.00 ± 05.53 | a | - | |
| 10.00 | 42.00 ± 07.05 | c | 90.00 ± 06.88 | a | - | |
| *p*-value | <0.0001 | | =0.8070 | | =0.0026 | |

*p*-value: probability value. Means followed by the same lowercase letter in the same row did not differ significantly according to the ranking of the generalized linear model with the binomial distribution. - Data were not considered in the analysis due to the low number of replications.

Larval mortality in the treatments with *M. albicans* showed a specific pattern (Figure 2). The 5% and 10% concentrations began to take effect only on the 4th day of exposure to the extract, and mortality peaked on Days 4 and 5. In contrast, the 1% concentration had an effect starting on the 2nd day, when there was also a peak in mortality.

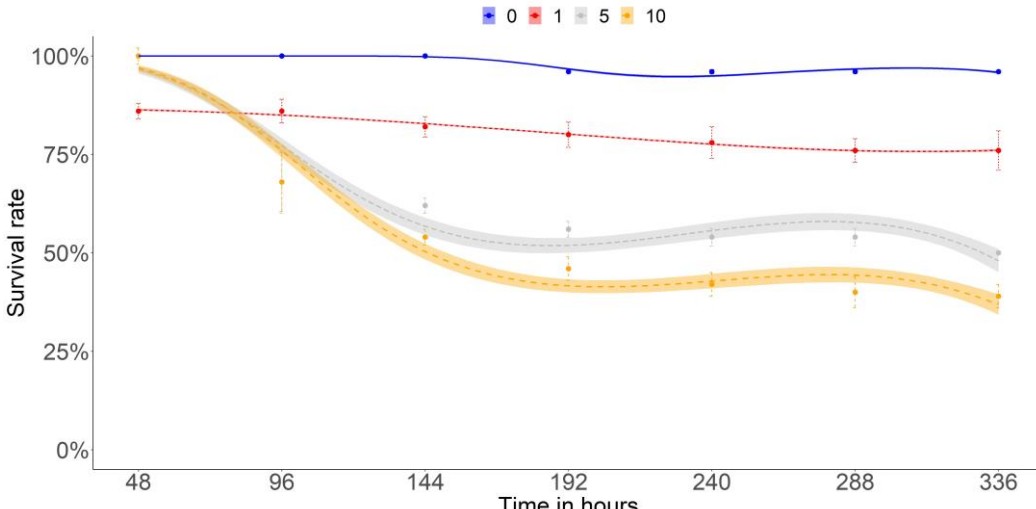

**Figure 2.** Larval survival every 48 h after exposure of *P. xylostella* larvae to botanical extracts of *M. albicans* at concentrations of 1%, 5%, and 10% and to the control.

Due to the high larval mortality at the extract concentrations of 5% and 10%, the number of individuals in the adult stage was not enough to reach the minimum number of replicates for statistical analysis. Thus, the treatments evaluated for the adult-stage parameters (adult longevity and egg survival) were the control and 1% botanical extract concentration treatments.

The longevity of females was reduced at the extract concentration of 1% ($\chi^2 = 10.94$, DF = 3, $p = 0.0155$). It was observed that females in this treatment lived half the time as those in the control (Table 3).

**Table 3.** Longevity (mean $\pm$ SE) of the adult stage of *P. xylostella* males and females exposed to different *M. albicans* extract concentrations.

| Concentration (%) | Female (Days) | | Male (Days) | |
|---|---|---|---|---|
| Control | 16.10 $\pm$ 01.21 | a | 20.90 $\pm$ 1.48 | a |
| 1.00 | 7.70 $\pm$ 01.85 | b | 18.00 $\pm$ 2.46 | a |
| *p*-value | =0.0155 | | =0.9673 | |

*p*-value: probability value. Means followed by the same lowercase letter in the column did not differ significantly based on the parametric ranking of the Gaussian model.

### 3.2. Free-Choice Feeding Preference Test with P. xylostella

The botanical extracts discouraged foliar consumption and were classified as feeding deterrents at concentrations of 10% (FPI = 0.299), 5% (FPI = 0.661), and 1% (FPI = 0.807) (Table 4). In addition, the 10% concentration also significantly reduced the consumed leaf area of the treated discs compared with the control (W = 91; *p* = 0.002) (Table 4).

The larvae exposed to the 10% concentration consumed almost 4.6 times more than the leaf area of the control, while this value was up to 1.57 times for the 5% and 1% concentrations. Furthermore, it is interesting to note that most of the *P. xylostella* larvae chose not to consume the discs treated with the extract, consuming only one of the discs offered with distilled water; that is, of the 30 replicates performed for each concentration, in 19 (1% concentration), 22 (5% concentration), and 24 (10% concentration) of them, the

larvae did not consume the discs treated with the extract. Figure 3 shows one of the scans obtained from the replicates and illustrates a comparison between the leaf consumption of the discs treated with the 10% extract concentration (A), which was the extract that caused the highest repellency, and the discs treated with distilled water (B). Figure 3 exemplifies what happened in most of the replicates.

**Table 4.** Feeding preference index (FPI) of *Plutella xylostella* according to consumed leaf area of discs treated with distilled water (control) or botanical extracts of *M. albicans*.

| Concentration (%) | Consumed Leaf Area (cm²) | | *p*-Value | FPI | Classification |
|---|---|---|---|---|---|
| | Extract | Control | | | |
| 1.00 | 0.105 ± 0.02 Aa | 0.126 ± 0.04 Aa | =0.272 | 0.807 | Fagodeterrent |
| 5.00 | 0.082 ± 0.01 Aa | 0.129 ± 0.03 Aa | =0.103 | 0.661 | Fagodeterrent |
| 10.00 | 0.043 ± 0.02 Ba | 0.201 ± 0.04 Aa | =0.002 | 0.299 | Fagodeterrent |
| *p*-value | =0.362 | =0.272 | | | |

Means followed by the same uppercase letter in the same row did not differ significantly based on the Mann–Whitney test at 5% probability. Means followed by the same lowercase letter in the column did not differ significantly based on the Kruskal–Wallis test at 5% probability. *p*-Value: probability value.

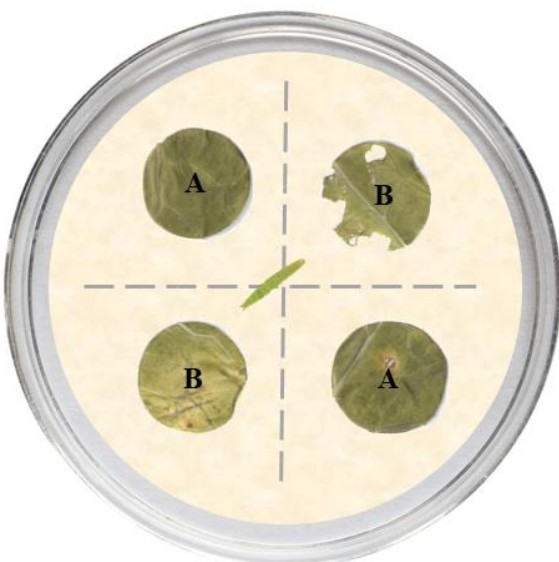

**Figure 3.** Comparison of consumed leaf area for *P. xylostella* between collard green discs treated with the botanical extract of *M. albicans* (10%) (A) and control treatment (B).

*3.3. Free-Choice Oviposition Preference Test with P. xylostella*

The botanical extracts discouraged females from laying eggs and were classified as deterrents at concentrations of 10% (FPI = 0.114), 5% (FPI = 0.668), and 1% (FPI = 0.684) (Table 5). The number of eggs on the treated discs was lower than that on the control at concentrations of 10% (W = 94; *p* = 0.0004), 5% (W = 86; *p* = 0.004), and 1% (W = 89; *p* = 0.0004). The egg count gradually decreased as the concentrations increased; however, the number of eggs in the 10% concentration discs was significantly lower than those in the discs at the other concentrations ($\chi^2$ = 17.62; DF = 2; *p* = 0.0001) (Table 5).

From the first day of observation, the adults preferred collard green discs without the extract for laying (Figure 4). Females exposed to the extracts at 10% concentration had a reduced oviposition period (last egg laying at 120 h) compared with those exposed to the concentrations of 5% and 1% (last egg laying at 240 h) (Figure 4). In general, females exposed to the 10% concentration suffered a premature death, which was not observed for males.

**Table 5.** Oviposition suppression index (OSI) of *Plutella xylostella* according to the number of eggs deposited on discs treated with distilled water (control) and with botanical extracts of *Miconia albicans*.

| Concentration (%) | Average Number of Eggs | | *p*-Value | OSI | Classification |
|---|---|---|---|---|---|
| | **Extract** | **Control** | | | |
| 1.00 | 83.60 ± 10.19 Ba | 160.80 ± 14.78 Aa | =0.001 | 0.684 | Deterrent |
| 5.00 | 54.50 ± 11.58 Ba | 108.70 ± 12.04 Aa | =0.004 | 0.668 | Deterrent |
| 10.00 | 05.70 ± 02.25 Bb | 94.70 ± 22.60 Aa | =0.0004 | 0.114 | Deterrent |
| *p*-value | 0.0001 | 0.0563 | | | |

Means followed by the same lowercase letter (in the column) and same uppercase letter (in the same row) did not differ significantly in the contrasts from the quasipoisson model at 5% probability. *p*-Value: probability value.

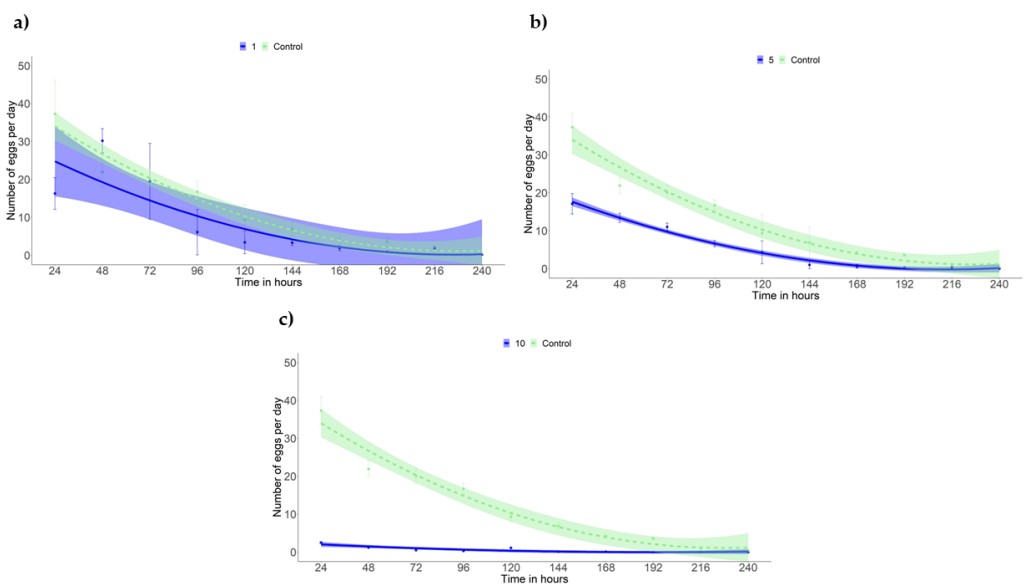

**Figure 4.** Number of eggs laid per day. Comparison between control (distilled water) and *M. albicans* extracts at concentrations of 1% (**a**), 5% (**b**), and 10% (**c**).

*3.4. Summary of the Lethal and Sublethal Effects of M. albicans*

Based on the results obtained, we compiled a flowchart of the lethal and sublethal effects of *M. albicans* on *P. xylostella*. As we observed that the effects were similar for all extract concentrations, varying only in intensity, they were divided into two groups.

Group I consisted of larvae that ingested botanical extracts (bioactivity experiment). The results revealed dysregulated larval duration (increase and decrease) and showed decreases in adult female longevity and larval mortality.

Group II consisted of individuals (larvae and adults) exposed to two substrate options: one with botanical extracts and the other with distilled water (preference experiments). The results were feeding deterrence, oviposition deterrence, and death of adult females (at 10% concentration).

All the effects listed were organized in a flowchart, simulating the action of the botanical extracts in the field (Figure 5).

*3.5. Chemical Analysis*

Phenolic compounds, flavonoids, and tannins were found in the botanical extract, increasing proportionally with the extract concentration (1%, 5%, and 10%) (Table 6). The same occurred with the antioxidant potential. Thus, the 10% concentration showed the highest number of chemical compounds, as well as the highest antioxidant potential (Table 6). The yield for all extracts was similar for all concentrations.

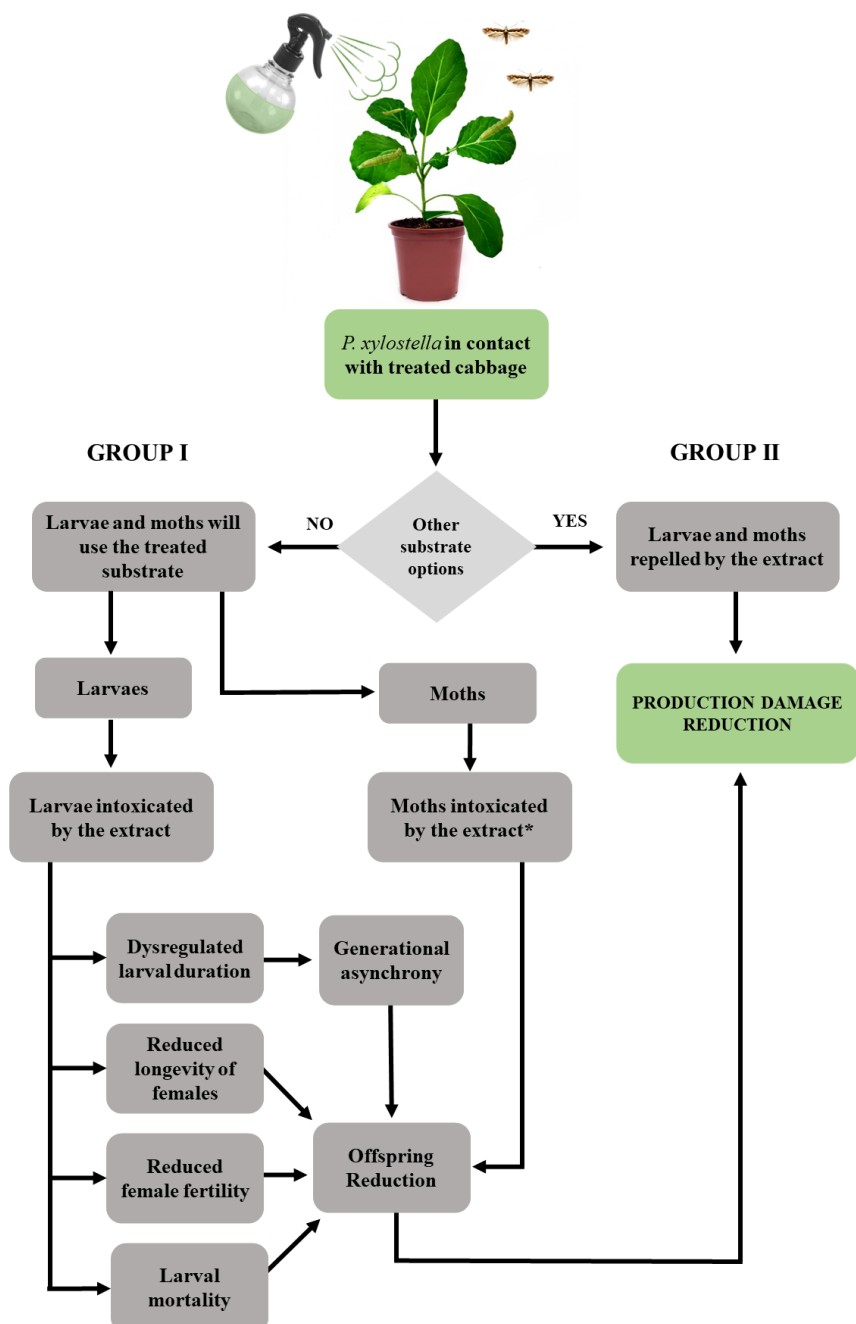

**Figure 5.** Flowchart of the action of aqueous extracts of *M. albicans* (1%, 5%, and 10%) on the life cycle of *P. xylostella*. * Effect only occurred at the 10% concentration.

**Table 6.** Data on phenolic compounds, flavonoids, tannins, antioxidant potential, and yield of botanical extracts of *Miconia albicans*.

| Sample | 1.00% | 5.00% | 10.00% |
|---|---|---|---|
| Phenolic compounds (mg AGE mL$^{-1}$ ± DP) | 30.93 ± 1.04 | 150.47 ± 1.99 | 296.87 ± 5.22 |
| Flavonoids (mg RE mL$^{-1}$ ± DP) | 14.37 ± 0.32 | 70.34 ± 0.57 | 139.46 ± 3.11 |
| Tannins (mg ATE mL$^{-1}$ ± DP) | 2.39 ± 0.07 | 12.02 ± 0.11 | 24.67 ± 0.13 |
| Antioxidant potential (% ± DP) | 5.61 ± 0.04 | 24.27 ± 0.15 | 49.48 ± 0.12 |
| Extracts' yield | 11.3% | 10.9% | 10.7% |

## 4. Discussion

This is the first study to evaluate the insecticidal potential of *M. albicans*. We found that the different concentrations of the aqueous extracts of the plant had antibiosis and antixenosis effects on the Diamondback moth, *P. xylostella*. The treatments caused larval mortality to feeding deterrence, which gradually increased according to the extract concentration.

The phytochemical analysis identified phenolic compounds, flavonoids, and tannins. Cunha et al. [53] performed a literature review on the main chemical constituents found in the genus *Miconia* spp., which corroborates the findings of the present study. Flavonoids and tannins function as a defense against attacks by plant pathogens and insect pests, presenting larvicidal and repellent effects [54–59]. Thus, the results of this study can be attributed to the presence of compounds from these groups. In addition, the number of chemical compounds varied proportionally to the extract concentration, which may explain the greater reductions in leaf consumption, oviposition, and larval survival for the 10% extract.

The bioactivity experiment showed three main outcomes: decreased larval survival, decreased adult hatching success, and dysregulation of larval duration.

The larvae that fed on the 10% treatment exhibited a mortality rate of up to 58%. Both flavonoids and tannins are compounds that can harm the development of insects after ingestion. Flavonoids can generate free radical species, which result in cellular toxicity [60,61], while tannins reduce enzyme movement and the existence of proteins [62], causing a nutritional obstruction. It is possible that larval digestibility was affected, leading to death over time.

An extended larval duration was observed for the highest concentrations, while it was reduced for 1% concentrations. This type of situation may cause asynchrony between the populations that consumed the extract and those that did not, causing reproduction among inbred individuals. The proportion of the compounds found in each concentration was different (Table 6), and the number of substances had different effects on the biology of insects, being lethal or sublethal. Once the lowest concentration was not lethal, sublethal effects were assessed in their populations, such as the extended larval duration. For the larvae fed with 1% concentrations, the proportion of compounds was not enough to kill most of the larvae, but it affected their development and capacity to generate offspring, as observed in Table 2.

The surviving female population showed reduced hatching success and longevity. The food consumed in the larval stage directly affects insect development and can probably reduce the number of ovarioles [63]. Therefore, the quality of the food ingested at this stage may have reduced the number of eggs per moth in the adult stage [64]. In other words, the extracts, even when nonlethal, caused damage to the pest's development and their ability to produce offspring.

Studies on the repellent action of flavonoids can be found in the literature [65–67]. The present study confirmed this type of sublethal effect of *M. albicans* on *P. xylostella*, pointing to both feeding and oviposition deterrence.

Of the larvae subjected to the feeding preference experiment, 96.66% consumed only the discs treated with aqueous extracts, which is important for producers, as the fact that the larvae did not give test bites in all discs could help them to avoid damage to commercial collard greens (Figure 3). Feeding suppression is an important mechanism because, in addition to reducing the damage caused by the insect, it prevents the larvae from continuing to feed on a substrate where coverage by the insecticide was irregular [68].

Likewise, all extract concentrations suppressed oviposition in *P. xylostella*, reducing the number of eggs laid on treated substrates by 94.72% for the highest concentration (10%) (Table 5, Figure 4). At the same time, it is interesting to highlight that the ratio of the eggs counted on treated and untreated collard green discs was different for each concentration. While at the 1% and 5% concentrations, this proportion was 34.2% and 33.9%, respectively, the extracts at 10% concentration reduced this proportion to 17.5%.

The presence of phytochemicals and their perception by insects may affect the occurrence or lack of oviposition [53]. Previous studies have identified that volatile compounds and their presence may alter *P. xylostella's* time of recognition of the substrates suitable for oviposition as well as laying behavior [69,70]. Thus, the presence of flavonoids and tannins could explain the results obtained, as already observed in the literature [9]. It is interesting to highlight that the oviposition period of the adults exposed to the 10% concentration was reduced. The females definitely stopped laying eggs after Day 7, which suggests that the compounds in the extract not only repelled the moths but also negatively affected their performance.

Figure 5 shows that *P. xylostella* was affected in at least three different ways, covering both the mature and immature stages of *P. xylostella*, and it could diversify the *M. albicans extract* mode of action in the field. The treatments repelled the larvae and adults in their search for a quality substrate, eliminated the larvae that fed on the treatments, and reduced the hatching success of surviving larvae.

Nevertheless, the *M. albicans* extract could affect other aspects of *P. xylostella* biology, such as its reaction to external factors (predators, parasites, parasitoids, and abiotic factors) after ingestion or topical application, which should be evaluated in future studies.

## 5. Conclusions

All concentrations of the aqueous extract of *M. albicans* were classified as feeding and oviposition deterrents. There was a reduction in larval survival and female hatching success after exposure to the extract in the juvenile stage. Among the concentrations used, the 10% treatment showed the best results. In addition, the phytochemical analysis showed that the plant extracts contained phenolic compounds, flavonoids, and tannins, substances with the reported ability to repel, inhibit, and intoxicate insects. This is the first study evaluating the insecticidal effect of *M. albicans*, and the results show that this plant has the potential to control *P. xylostella* and may be an accessible and economically viable tool, especially for small farmers.

**Author Contributions:** Conceptualization, I.M.P.M.P., S.A.d.S. and R.M.M.; methodology, I.M.P.M.P., S.A.d.S., C.A.L.C. and R.M.M.; formal analysis, I.M.P.M.P., J.K.d.S.P. and J.B.M.; investigation, I.M.P.M.P. and S.A.d.S.; writing—original draft preparation, I.M.P.M.P.; writing—review and editing, I.M.P.M.P., S.A.d.S., C.A.G.F. and R.M.M.; supervision, S.A.d.S. and R.M.M. All authors have read and agreed to the published version of the manuscript.

**Funding:** This study was funded by the Development of Education, Science and Technology of Mato Grosso do Sul (FUNDECT) for the resources provided under No. 71/711.130/2018.

**Institutional Review Board Statement:** Not applicable.

**Informed Consent Statement:** Not applicable.

**Data Availability Statement:** The data presented in this study are available on request from the corresponding author.

**Acknowledgments:** We would like to thank the National Council for Scientific and Technological Development, Brazil (CNPq) for providing a scholarship to the first author. This study was supported by the Federal University of Grande Dourados, Dourados, MS.

**Conflicts of Interest:** The authors declare no conflict of interest.

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
