# Peer review of "Leaf Extracts of Miconia albicans (Sw.) Triana (Melastomataceae) Prevent the Feeding and Oviposition of Plutella xylostella (Linnaeus, 1758) (Lepidoptera: Plutellidae)"

_agronomy, doi:10.3390/agronomy13030890_

Round 1

Reviewer 1 Report

In my opinion, the manuscript is interesting and the data are worth publishing. The average originality is not an objection - standard, classical methods are still crucial, especially when new substances/extracts are tested. 

However, I have some remarks:

Lines 73-75: "According to the Arthropod Pesticide Resistance Database, P. xylostella is currently resistant to 101 active ingredients" - I suggest "P. xylostella populations have been reported to be resistant to 101...". The database refers to tested populations and it does not mean that all populations (whole species) are resistant to all 101 active ingredients.

Lines 97-99: Is this a new technique or a standard one? If a standard one - please give a citation.  

The term "fertility" is used when the authors show the number of hatched larvae. I would suggest "hatching success". The observation does not refer to fertilization only (i.e. formation of zygote) but also to the development of the organism in the ovum. 

Line 124 - were the adults fed? 

Some subchapters are not self-descriptive (2.5, 2.6, 2.7, 2.8). For example "phenolic content" does not inform where was it measured. In a plant/an extract/a solution? That is not clear to the reader. Next, "To 0.1 mL of sample ..." - what is "a sample"? Similarly in 2.6 and 2.7. 

Line 228: "The blank used was distilled water" - why not 2% aluminum chloride in methanol? Please explain. 

I suggest beginning results from mortality. Usually, as the crucial endpoint, mortality is presented first. 

The titles of the tables must be corrected. Both tables and figures must be self-explanatory. So: "Durations (mean ± SE) and nonparametric ranking of the larval and pupal stages exposed to different extract concentrations" must be corrected to: "Durations (mean ± SE) and nonparametric ranking of the larval and pupal stages of P. xylostella exposed to different M. albicans extract concentrations" and so on. Please give units in tables! The reader does not know if the data are in days/hours/weeks etc. Also, explain what the letters in brackets mean, under the tables. 

Line 227: "it did not differ significantly from..." (add "significantly")

Lines 297-300: these results need a more precise description. What kind of malformations were observed? How many insects looked in the described way? Any ratio/percentage of malformed organisms? 

Table 4 - what are the values in the "Extract" and "Control" columns? 

Line 322: "while this value was up to 1.57 times for the 5% and 1% concentrations" - these data should be presented precisely, perhaps in a table. 

Figure 4 is a little bit enigmatic. First, why only two concentrations are presented? Next, only one leaf (of four) is affected. The right control leaf is not affected, too. This figure suggests that the insects did not eat exposed leaves et all, while Table 4 shows some feeding on exposed leaves. So, the pictures are not representative. 

Line 321: "... consumed almost 4.6 times the leaf..." - 4.6 times more or 4.6 times less?

Table 5. I suggest showing the ratio/percentage of eggs oviposited on tested and control leaves. In the case of 1% and 5%, they are almost the same (34.2% and 33.9%, respectively). This needs a comment in the discussion.

Line 350-351: "In general, females exposed to the 10% concentration suffered premature death, which was not observed for males." - Figure/table? 

Figure 5 - begin from the lowest concentration, as you did everywhere. 

Discussion should be developed. Is there a correlation between food consumption and growth/development/mortality/longevity? The authors also state that "Of the larvae subjected to the feeding preference experiment, 96.66% consumed only the disks treated with aqueous extracts" and "all extract concentrations suppressed oviposition in P. xylostella, reducing the(m) by 94.72% the number of eggs laid on treated substrates". No such data had been presented in the results. If they are based on any calculations presented there, the authors must mention that, referring to a proper table/figure. 

Lines 440-442: What is the significance of this observation and what could be the cause? Feeding? 

The authors do not discuss their non-linear results (which actually, are quite interesting". Why the results are not always linear? Is there any hormetic range here? In my opinion, the authors obtained many interesting results, which are not fully discussed.

Author Response

We appreciate the reviewer's valuable suggestions. We inform that the suggestions given have all been understood, some sentences have been altered as indicated by this reviewer, and other reviewers and information was added. Other questions will be clarified directly in the document attached below.

Reviewer 2 Report

  The experiments were well-designed to investigate the effects of Miconia albicans botanical extracts on the biological activity, feeding preference, and oviposition preference of Plutella xylostella. In addition, preliminary chemical analysis was conducted on the Miconia albicans botanical extracts. It is significant for the development of biopesticides.

The study is well performed and results are convincing.

However, several issues should be addressed before publishing the results.

1.       Relevant information about Plutella xylostella population should be provided in the method part.

2.       Error bars should be added to the mortality statistics in Figure 2.

3.       There is a significant difference in the presented larval size currently used in Figure 3. It is recommended to use larvae with similar body sizes

4.       "Error bars are missing from the oviposition data in Figure 5."

5.       The test for the lethal concentration (LC50) of the extract should be conducted, as this would better demonstrate the potential application of the extract."

Author Response

We appreciate the reviewer's valuable suggestions. We inform that the suggestions given have all been understood and some sentences have been altered as indicated by this reviewer and other reviewers and information was added. Other questions will be clarified directly in the document attached below.

Reviewer 3 Report

comments in the document

Author Response

(The authors gave the same response as above.)

Round 2

Reviewer 1 Report

The ms has been improved. Some minor remarks:

line 138 - 2.3 Effects of M. albicans extract on P. xylostella

line 210 - Chemical Analyzes of M. albicans extracts – “a”, not “A”.

Table 2: use “-“ instead of “***” for “Data were not considered in the analysis due to the low number of replications”

Author Response

We appreciate the reviewer's valuable suggestions. We inform that the suggestions given have all been understood, and the sentences were altered as indicated by this reviewer.

Reviewer 3 Report

The paper is good, 

Author Response

We appreciate the reviewer's valuable suggestions. We inform that the suggestions given have all been understood, some sentences have been altered as indicated by the reviewers, and the English spell was checked.
